# Studying genetic population structure to shed light on the demographic explosion of the rare species *Barbitistes vicetinus* (Orthoptera, Tettigoniidae)

Isabel Martinez-Sañudo[1]*, Corrado Perin[1], Giacomo Cavaletto[1], Giacomo Ortis[1], Paolo Fontana[2], Luca Mazzon[1]*

1 Department of Agronomy, Food, Natural resources, Animals and Environment (DAFNAE), University of Padua, Legnaro, PD, Italy, 2 Istituto Agrario San Michele all'Adige (IASMA) Research and Innovation Centre, Foundation Edmund Mach (FEM), San Michele all' Adige, Trento, TN, Italy

* isabel.martinez@unipd.it (IMS); lmazzon@unipd.it (LM)

**Data Availability Statement:** All sequence files are available from the NCBI-GenBank database (accession numbers: MW405351- MW405381).

## Abstract

Insect outbreaks usually involve important ecological and economic consequences for agriculture and forestry. The short-winged bush-cricket *Barbitistes vicetinus* Galvagni & Fontana, 1993 is a recently described species that was considered rare until ten years ago, when unexpected population outbreaks causing severe defoliations across forests and crops were observed in north-eastern Italy. A genetic approach was used to analyse the origin of outbreak populations. The analysis of two mitochondrial regions (Cytochrome Oxidase I and II and 12S rRNA-Control Region) of 130 samples from the two disjunct ranges (Euganean and Berici Hills) showed high values of haplotype diversity and revealed a high geographical structure among populations of the two ranges. The high genetic variability observed supports the native origin of this species. In addition, results suggest that unexpected outbreaks are not a consequence of a single or few pestiferous haplotypes but rather the source of outbreaks are local populations which have experienced an increase in each area. The recent outbreaks have probably appeared independently of the genetic haplotypes whereas environmental conditions could have affected the outbreak populations. These findings contribute to a growing understanding of the status and evolutionary history of the pest that would be useful for developing and implementing biological control strategies for example by maximizing efforts to locate native natural enemies.

## Introduction

Herbivorous insect outbreaks usually involve important ecological and economic consequences for agriculture and forestry. In most cases, it concerns the accidental introduction of an exotic species into a new geographical area due to international trade and human movement [1–3]. In this scenario, the species is not under biological control by natural enemies, leading to a quick spread of the population. Instead, the insect outbreak of an indigenous

**Funding:** This research was supported by the project DOR1881327/18 - University of Padua to Luca Mazzon. The funders had no role in study design, data collection and analysis, decision to publish, or preparation of the manuscript. There was no additional external funding received for this study.

**Competing interests:** The authors have declared that no competing interests exist.

species is less common but more enigmatic, due to some alterations of the biotic and abiotic components [4].

Until a decade ago, among bush-crickets belonging to the genus *Barbitistes*, (Orthoptera, Tettigoniidae), occasional outbreaks were reported only for *B. constrictus* Brunner von Wattenwyl, 1878 in conifer forests of central and eastern Europe, for *B. ocskayi* (Charpentier, 1850) in broadleaf forests of north-eastern Italy [5–7] and for *B. serricauda* (Fabricius, 1798) in vineyards of north Italy. In spring 2008, the first outbreak of the species *B. vicetinus* Galvagni & Fontana, 1993 was recorded in a restricted area of north-eastern Italy (Euganean Hills) [7]. Since then, within two disjunct distribution ranges, the Euganean Hills first and the Berici Hills later, the outbreak areas have progressively enlarged causing heavy damage to forests and neighboring crops (mainly grapes and olives). Severe defoliations have been recorded in the most serious infestations, rising to nearly 90% of canopy loss [8, 9]. Moreover, the outbreaks are also a source of annoyance to people living close to the attacked areas due to the tendency of bush-crickets to invade streets and gardens [7, 10] (S1 Fig).

Interestingly, the bush-cricket was first described just in 1993 as a rare endemic species of north-eastern Italy, being found only in small confined hilly areas [11]. From then on, all the records of *B. vicetinus* have been reported only in this area. Only since 2005 this species started to be more common in some localities and easy to be found also without bioacoustics technics. The recent discovery of the species (Galvagni & Fontana, 1993) and increase in these north-eastern areas affected by outbreaks during the past two decades has sparked a debate about the possible exotic origin of *B. vicetinus*. Recently, as a consequence of the outbreaks, some authors have thoroughly studied important aspects of the biology and ecology of *B. vicetinus* [7, 9, 12, 13] but no data regarding population genetics is so far available. Genetic studies based on the use of high-resolution DNA markers allow to examine the structure of insect populations, identify haplotypes, reconstruct current or past patterns of gene flow and provide information on the origin and expansion routes of the insect [14]. This information could be useful for improving the knowledge concerning the eruptive species, such as colonizing capacity, adaptability, behaviour and demographic history of a population. [15–17]. Several population genetics studies have been conducted with the aim of highlighting and discovering the sources and colonization routes of outbreak populations of alien pests [e.g. 18–23]. However, studies on the genetic structure of native pests have generally received less attention [4, 24].

Among the genetic tools to be exploited, maternally-inherited mtDNA is widely used since it is relatively conserved compared to some nuclear genes, and is thus suitable when searching for historical processes [25]. In this study, we used a combination of two mitochondrial markers, namely Cytochrome Oxidase I and II (COI-tRNALeu-COII) and 12S rRNA-Control Region (12S-CR) to study the genetic structure and demographic history of *B. vicetinus* populations from the two disjunct ranges where it currently outbreaks (Euganean Hills and Berici Hills, northern Italy). The COI and COII are considered slowly evolving genes of the mitochondrial protein-coding genes [26]. The non-coding control region is in contrast, the most variable segment in the maternally inherited mtDNA [27, 28] and its use in genetic analysis can alleviate biases of coding regions [29]. Both markers have been widely used to conduct population genetic surveys in Orthopteran species including the Tettigoniidae family [30–35].

By studying the genetic structure of the bush-cricket this study aims to answer three main questions:

i. is there any evidence for an alien origin of the bush-cricket?

ii. are *B. vicetinus* populations genetically distinct?

iii. are either of the recent outbreaks caused by a single outbreak population followed by a spread pattern or have they derived from multiple local populations?

This information can help understand the factors related to the outbreak events of *B. vicetinus* and could provide insight to assist management of this outbreak pests.

## Materials and methods

### Study area

The study area was located in north-eastern Italy (Veneto Region) and included the two disjunct ranges where outbreaks have occurred (Euganean Hills and the Berici Hills) and a third range where no outbreaks have been reported to date and the species is rare (Lessini Mountains). The three ranges are separated from each other by cultivated and inhabited alluvial plains devoid of woody vegetation (Fig 1B).

The Euganean Hills cover an elliptical area of approximately 180 km$^2$ and comprise approximately 100 hills of volcanic origin emerging from the alluvial sedimentary plain, with the highest elevation of approximately 600 m above sea level. They are characterised by numerous narrow and deep valleys, steep hills with different sun exposures and many microclimatic conditions that influence the vegetation. The annual mean temperature is approximately 12°C and precipitation ranges from 700 to 900 mm [36] even if they are extremely variable due to the inner geomorphological variability.

The Berici Hills are an isolated plateau situated on the southern Vicenza plain, they cover almost 160 km$^2$ with a maximum elevation of approximately 450 m above sea level [37]. The climate of the area is characterised by an annual rainfall of 958 mm, and average daily temperature of -1°C in January and 23°C in July [38].

The Lessini Mountains are a triangular-shaped tableland, which occupies some 800 km$^2$, at the transition between the Fore-Alps and the River Po Plain [39]. The mountain group reaches over 2000 m above sea level and is characterised by multiple valleys and long ridges that descend towards the plain.

### Insect sampling

Adults of *B. vicetinus* were collected, using a sweep net, throughout the spring of three successive years (2015, 2016, and 2017). Specimens from the north, centre and south part of both outbreak areas (the Euganean and Berici Hills) as well as from two sites on the Lessini Mountains were collected from bush and tree canopies on marginal side of the forests (Table 1). To avoid sampling relatives, bush-cricket specimens were collected by sweeping an area at least 100 m$^2$ at each sampling site. After capturing, samples were immediately kept in 95% ethanol and taken to the laboratory where they were morphologically identified and stored in individual vials at −20°C until DNA extraction.

### DNA extraction, PCR amplification and sequencing

The genomic DNA was extracted from tissue samples taken from the hind femora of each specimen separately, according to a previously described salting-out protocol [40]. Two mitochondrial DNA regions were chosen for amplification: a fragment including the 3' region of Cytochrome Oxidase I, tRNA-Leu and the 5' region of Cytochrome Oxidase II (COI-tRNALeu-COII) and a fragment of the 12S rRNA and the Control Region (12S-CR). Two specific primers were designed for amplifying the COI-tRNALeu-COII fragment: Vicetinus_P7F (5'-ACCTGT TCTTGCAGGAGC-3') and Vicetinus_P4R (5'-TCCACAGATTTCTGAACATTG-3'). The universal primers SR-J14610 (5'-ATAATAGGGTATCTAATCCTAGT-3') and T1-N18

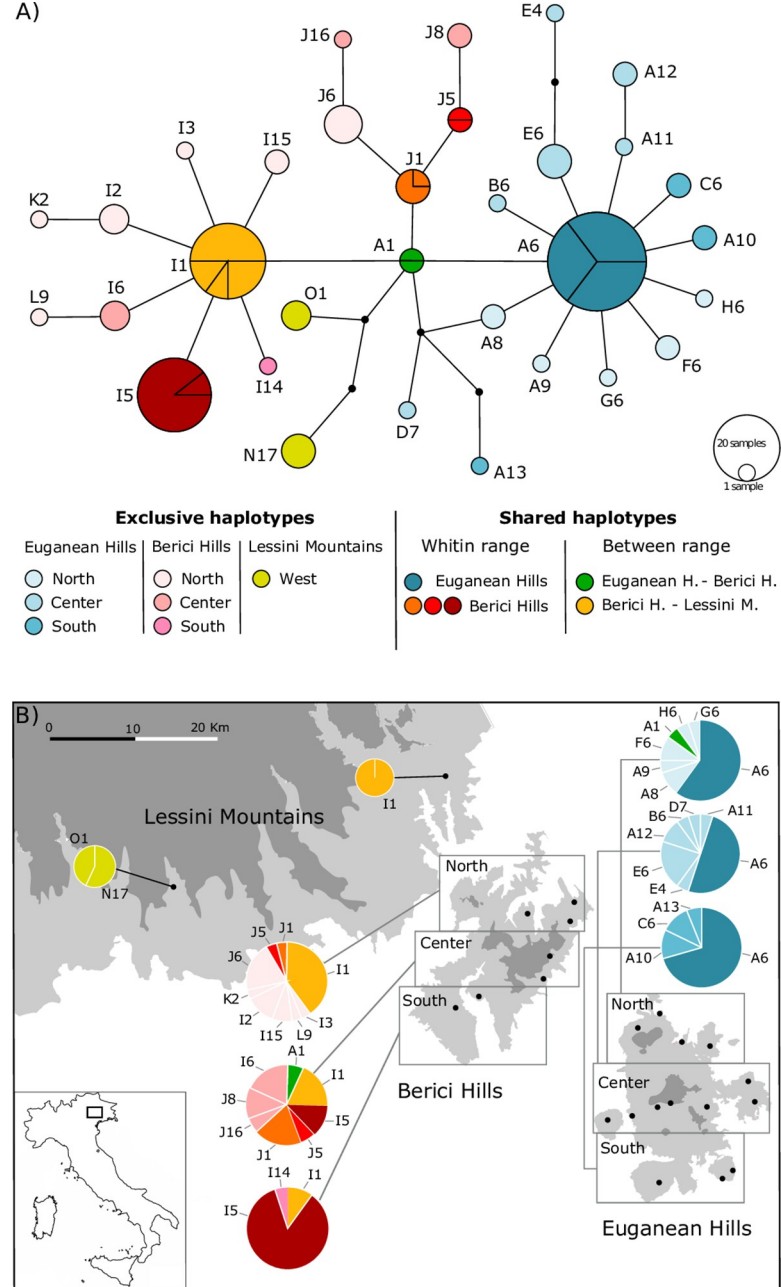

**Fig 1. Parsimony network and geographic distribution of *B. vicetinus* haplotypes.** A) Haplotypes network based on the combined dataset (COI-tRNALeu-COII and 12S-CR). Each haplotype is represented by a circle, and the area of the circle is proportional to its frequency. Lines within haplotypes circles indicate the proportions shared between collection areas. The colours represent differences in geographic distribution, and small black dots symbolize missing intermediate or unsampled haplotypes. Codes indicate the haplotype ID, reported in Table 1. B) Map showing the proportional geographic distribution of bush-cricket haplotypes across sampled populations. Map republished from [62] under a CC BY license, with permission from [Regione del Veneto–L.R. n. 28/76 –Formazione della Carta Tecnica Regionale], original copyright [2020].

(5'-CTCTATCAARRTAAYCCTTT-3') [41] were used to amplify the 12S-CR fragment. Amplifications were performed in 20 μl reactions (1x PCR Go Taq Flexi buffer—Promega, 2.5 mM MgCl2, 0.1 mM dNTPs, 0.5 μM for each primer, 0.5 U of Taq polymerase—Promega, 2 μl

Table 1. Collection sites of *B. vicetinus* populations analysed and descriptive statistics of each population.

| | | Site | | | | | N. haplotypes | H | π (%) |
|---|---|---|---|---|---|---|---|---|---|
| | | ID | Latitude | Longitude | N. samples | Haplotypes | | | |
| Euganean Hills | North | 1 | 45˚20'46.4"N | 11˚44'4.2"E | 1 | A6(1) | 7 | 0.64 +/- 0.12 | 0.04 |
| | | 2 | 45˚22'30.6"N | 11˚40'38.7"E | 10 | A6(7), F6(2), A9(1) | | | |
| | | 3 | 45˚21'54.3"N | 11˚38'52.7"E | 8 | A1(1), A6(4), A8(2), H6(1) | | | |
| | | 4 | 45˚20'59.9"N | 11˚41'52.8"E | 1 | G6(1) | | | |
| | Center | 5 | 45˚19'14.8"N | 11˚45'57.8"E | 1 | A6(1) | 7 | 0.72 +/- 0.09 | 0.09 |
| | | 6 | 45˚17'38.1"N | 11˚37'22.5"E | 2 | A6(2) | | | |
| | | 7 | 45˚18'14.6"N | 11˚40'26.8"E | 1 | D7(1) | | | |
| | | 8 | 45˚18'9.07"N | 11˚40'51.7"E | 9 | A6(1), A11(1), A12 (2), B6(1), E4(1), E6(3) | | | |
| | | 9 | 45˚18'12.4"N | 11˚43'46.6"E | 3 | E6(1), A6(2) | | | |
| | | 10 | 45˚17'45.6"N | 11˚39'35.3"E | 3 | A6(3) | | | |
| | | 11 | 45˚18'9.7"N | 11˚46'48.5"E | 1 | A6(1) | | | |
| | South | 12 | 45˚14'53.8"N | 11˚44'39.5"E | 12 | A6(8), A10 (2), C6(2) | 4 | 0.50 +/- 0.13 | 0.05 |
| | | 13 | 45˚14'41.1"N | 11˚40'6.2"E | 2 | A6(1), A13(1) | | | |
| | | 14 | 45˚14'55.1"N | 11˚44'39.3"E | 3 | A6(3) | | | |
| Berici Hills | North | 15 | 45˚28'20.7"N | 11˚35'27.5"E | 16 | I1(10), I2(1), I3(1), I15(2), J6(1), L9(1) | 9 | 0.80 +/- 0.06 | 0.10 |
| | | 16 | 45˚26'41.2"N | 11˚34'42.1"E | 1 | J6(1) | | | |
| | | 17 | 45˚27'59.7"N | 11˚32'52.9"E | 8 | I2(2), J1(1), J5(1), J6(3), K2(1) | | | |
| | Center | 18 | 45˚25'31.5"N | 11˚33'02.8"E | 1 | I5(1) | 8 | 0.90 +/- 0.04 | 0.13 |
| | | 19 | 45˚24'00.3"N | 11˚31'39.7"E | 15 | A1(1), I1(3), I5(1), I6(3), J1(3), J5(1), J8(2), J16(1) | | | |
| | South | 20 | 45˚25'14.1"N | 11˚26'59.0"E | 17 | I5(17) | 3 | 0.28 +/- 0.12 | 0.02 |
| | | 21 | 45˚24'54.8"N | 11˚28'59.9"E | 3 | I1(2), I14(1) | | | |
| Lessini Mountains | | 22 | 45˚29'12.3"N | 11˚02'14.6"E | 7 | N17(4), O1(3) | 3 | 0.71+/- 0.07 | 0.24 |
| | | 23 | 45˚36'14"N | 11˚25'18"E | 5 | I1(5) | | | |

DNA template). Thermal cycling conditions for the fragment including the COI-tRNALeu-COII were 5 min at 96˚C followed by 35 cycles at 96˚C for 1 min, 56˚C for 1 min, 72˚C for 1:30 min, and a final extension of 72˚C for 5 min. For the 12S-CR amplification, the thermal profile followed the conditions described by Eweleit et al. [33] consisting of 2 min at 92˚C followed by 35 cycles with a denaturation step of 92˚C for 2 min, an annealing step of 52˚C for 30 sec, and extension step of 60˚C for 3 min, with a final extension of 72˚C for 7 min.

PCR products were checked through electrophoresis on 1.0% agarose gels stained with SYBR® (Invitrogen), purified using Exonuclease and Antarctic Phosphatase (GE Healthcare) and sequenced at the BMR Genomics Service (Padua, Italy). Primers used for amplification were also used for sequencing.

## Data analysis

DNA sequence chromatograms were quality checked, manually corrected when necessary, and aligned using MEGA X [42]. Low-quality regions found at the beginning and end of each sequence were trimmed, while low-quality sequences were not included in the analysis.

Haplotype and nucleotide diversity, as well as the pairwise genetic distances between populations, were calculated with Arlequin 3.5 [43] using a Kimura 2-parameters model. The presence of population differentiation was also tested by conducting exact tests of population differentiation with 100,000 steps in Markov chain, with 10,000 dememorization steps. To compare the partition of genetic variability among sampled populations, an analysis of molecular variance (AMOVA) [44] was performed using Arlequin.

A haplotype parsimony network with a probability cut-off of 95%, was reconstructed using the TCS 1.21 software [45] and PopART 1.7 [46] and used for depicting the geographical relationships among haplotypes. Ambiguous connections (loops) were resolved using approaches from coalescent theory based on three criteria: frequency, network location and geography [47, 48].

The study of the past demographic history of the species was inferred using Arlequin 3.5 through the Tajima's D and Fu's Fs tests [49, 50] and the mismatch distributions of the pairwise genetic differences [51]. Populations at demographic equilibrium or decreasing in size should provide significant positive D and Fs values with a multimodal distribution of pairwise differences, whereas populations that have undergone a sudden demographic expansion usually show significant negative D and Fs values with a unimodal distribution [51, 52]. The sudden expansion model was tested through analysis of the sum of square deviations (SSD) and raggedness index (r) representing the modality of distribution, obtaining the corresponding P values with a parametric bootstrap approach (10,000 replicates).

For those populations that did not deviate from sudden expansion (p > 0.05) the time since expansion was calculated considering the relationship $\tau = 2^*u^*t$ (where $\tau$ = age of expansion measured in units of mutational time, u = mutation rate per sequence and per generation, and t = number of generations since the expansion; [53]). The expansion time was then obtained by dividing the estimate of $\tau$ by the product of the sequence length in base pairs and the mutation rate per nucleotide (twice the per-lineage substitution rate; $u = 2 \mu k$) in percentage per year. Two substitution rates were used: the classical rate of 2.3% divergence per Myr [54] frequently used in Orthoptera [55–59] and the global mtDNA rate of 2.6% divergence per Myr proposed by Papadopoulou et al. [60].

## Results

### Sampling

A total of 152 individuals were collected from the three natural ranges of *B. vicetinus* (Euganean Hills, Berici Hills and Lessini Mountains). Unfortunately, only 12 specimens were collected from the Lessini Mountains due to the rarity of the species in this range (Tab. 1).

### Data analysis

The two fragments of the mitochondrial genes, COI-tRNALeu-COII and 12S-CR were successfully amplified and sequenced in 130 samples, with an average of 19.6 specimens for each of the three areas within the two main ranges (north, centre, and south of both the Euganean and Berici Hills). After quality assessment and trimming the sequences, high-quality sequences 799 bp long for the COI-tRNALeu-COII and 898 bp long for the 12S-CR fragment were obtained. Sequences of the COI-tRNALeu-COII fragment were translated with Transeq (EMBOSS: http://www.ebi.ac.uk/Tools/emboss/transeq/index.html) to exclude the presence of any nuclear mitochondrial pseudogenes. This bioinformatics tool translates nucleic acid sequences to their corresponding peptide sequences and identifies stop codons. Since these pseudogenes (NUMTs) are characterized by the accumulation of in-frame stop codons and indels [61] the absence of stop codons in the protein sequence can allow to exclude presence of NUMTs.

A total of 13 variable sites, including 8 parsimony informative sites and 5 singleton sites, were identified by the alignment of COI-tRNALeu-COII sequences while the 12S-CR alignment showed 6 parsimony informative sites and 7 singleton sites (13 variable sites in total). Sequences of each haplotype of COI-tRNALeu-COII and 12S-CR obtained in this study are available through GenBank accession numbers MW405351- MW405381.

Partition homogeneity test confirmed that COI-tRNALeu-COII and 12S-CR fragments bear a homogeneous signal (P = 0.28), allowing data to be pooled for further analyses. A final dataset including 130 concatenated sequences of 1697 bp was obtained. Further analyses were conducted considering the combined dataset.

## Genetic variability and population structure

The diversity indexes for the concatenated dataset ranged between 0.28 and 0.90 for the haplotype diversity (H) and between 0.02 and 0.24 for the nucleotide diversity (π). Among populations from both the main outbreak areas, those from the north and the centre part of the Berici Hills showed the highest H and π values (H = 0.92, π = 0.13 and H = 0.80, π = 0.10 respectively) while populations from the south part of the Berici Hills showed the lowest variability. The distribution of haplotypes among all populations analysed and other summary statistics are shown in Table 1.

The presence of population differentiation was confirmed by the tests of population differentiation (P>0.001). When *B. vicetinus* genotypes were grouped based on the six main collection sites (north, centre, and south of both Euganean and Berici Hills), the locus-by-locus AMOVA revealed a significant geographic structure between populations of the two main disjunct ranges (P = 0.0014) (Euganean vs Berici Hills). To avoid bias as a result of the small sampling size of Lessini Mountains, populations from this area were not included in this analysis. The analysis showed that 40% of the variation was explained by differences among groups (Euganean and Berici Hills), whereas about 48% of genetic variation was explained within populations (Table 2).

## TCS Network

TCS Network of the combined dataset revealed the presence of 32 haplotypes of which 15 were exclusive to samples from the Euganean Hills, 13 were exclusive to the Berici Hills and 2 regarded samples only from the Lessini Mountains (Fig 1). Only 2 haplotypes were shared by samples from different ranges: A1 included samples from the Euganean and Berici Hills, and J1 samples from both the Berici Hills and Lessini Mountains. The network evinced a geographical separation among haplotypes, with samples from the Euganean Hills showing a star-like pattern and samples from the Berici Hills connected among them (Fig 1A).

Going deeper into the network's characteristics, the most common haplotype, A6, included 34 samples coming from all three parts of the Euganean Hills (north, centre and south). Ten rare haplotypes including samples exclusively from the Euganean Hills were connected, separated by only one mutational step, to A6 in a star-shape. Five of them (A9, A11, B6, G6, and H6) were represented by only one sequence, six (C6, A10, F6, A8) by two sequences, and one (E6) by 4 sequences. In addition, haplotype A1, which included samples from both the Euganean and Berici Hills, was also connected to A6 separated from it by only one mutational step.

The second most common haplotype, I1, was composed of 20 sequences:15 belonged to samples from the three geographical populations of Berici Hills (north, centre and south)

**Table 2. Analysis of molecular variance AMOVA for the combined data set (COI-tRNALeu-COII and 12S-CR).**

| Structure | Source of variation | Variance % | Fixation indices | P-value |
|---|---|---|---|---|
| **Grouping by geographical range (Euganean and Berici Hills)** | Among groups | 40.32 | $F_{CT} = 0.403$ | 0.0014 |
| | Among populations within groups | 11.75 | $F_{SC} = 0.197$ | <0.001 |
| | Within populations | 47.92 | $F_{ST} = 0.520$ | <0.001 |

Amova was calculated among populations of *Barbitistes vicetinus* divided according to the outbreak areas.

while 5 were represented by samples from site 23 (Lessini Mountains). This haplotype (I1) was separated from the most dominant haplotype (A6) by two mutational steps. A star-like pattern could also be observed in this part of the network, with haplotype I1 surrounded by nine haplotypes, eight of which (I2, I3, I5, I6, I14, I15, K2 and L9) exclusively represented by samples from the Berici Hills. Among them, haplotype I5 was shared by 19 samples from the southern and central populations of the Berici Hills while haplotypes I2, I3, I15, K2 and L9 were rare (3 sequences each at most) and included only samples for the north part of the Berici Hills. Two haplotypes, O1 and N17, were exclusive to samples from site 22 (Lessini Mountains) the furthest site from the outbreak areas (Fig 1).

The parsimony networks of the single markers (COI-tRNALeu-COII and 12S-CR) showed a similar pattern, with most haplotypes including exclusively samples from only one disjunct distribution range (Euganean Hills or Berici Hills or Lessini Mountains) (S2 and S3 Figs).

### Past demographic events

Tajima's D and Fu's Fs tests were applied in populations from the Euganean and Berici Hills in order to check for past demographic events. The null hypothesis of neutrality was rejected in populations from the Euganean Hills (D = -2.05; P = 0.02 and Fs = -12.51; P < 0.001), suggesting a past population expansion after a period of low effective sample size (Table 3). Berici populations showed only significant negative Fu's Fs value (Fs = -6.62; P = 0.004), suggesting that *B. vicetinus* populations of this hilly area did not conform to the theory of neutral evolution (Table 3).

The mismatch distribution plots of both ranges were smooth and unimodal, revealing that these populations were undergoing population expansion (S4 Fig). They were characterised by the following observed means: 1.25 and 1.91 for the Euganean and Berici Hills, respectively. Moreover, for both the Berici and Euganean populations, the computed SSD and raggedness index values did not reject a sudden expansion model (S4 Fig), and in particular raggedness values were low (Table 3).

Estimation of the expansion time showed that populations from the Euganean Hills lineage started to expand about 10,700 years ago (with a 2.6% substitution rate) and 12,400 years ago (with a 2.3% substitution rate) (Table 3). Populations from the Berici Hills probably expanded from about 24,000 to 27,600 years ago with 2.6% and 2.3% substitution rates, respectively (Table 3).

## Discussion

This study presents the first population genetic analysis of the species *B. vicetinus* in its outbreak areas and reveals the presence of a high geographical structuring among populations of the two outbreak ranges analysed (Euganean and Berici Hills).

**Table 3. Statistic summary of the past demographic events analysis of *Barbitistes vicetinus* populations.**

| Area | Tajima's D | Fu's Fs | SSD | r | τ (confidence interval 95%) | Expansion time (ka) 2,3% subst.rate | Expansion time (ka) 2,6% subst.rate |
|---|---|---|---|---|---|---|---|
| Euganean Hills | -2 .05* | -12 .51* | 0 .009 | 0.069 | 0.96 (0.34–1.67) | 12.4 (4.4–21.6) | 10.7 (3.8–18.7) |
| Berici Hills | -0 .23 | -6 .62* | 0 .002 | 0.030 | 2.14 (0.54–3.62) | 27.6 (7–46.7) | 24 (6.1–40.3) |

Tajima's (D) and Fu's neutrality test (Fs) mismatch distribution analysis under a sudden expansion model and time since expansion calculated for mitochondrial populations of the Euganean and Berici Hills considering the combined data base. SSD: sums of squared deviations; r: raggedness index, τ: age of expansion measured in units of mutational time. Expansion time shows as 1,000 years ago (ka).

* Significant at p < 0.05.

Populations from both ranges showed high values of haplotype diversity, a typical characteristic of ancestral populations [63–65], supporting that *B. vicetinus* is a native species. Conversely, populations of invasive alien species are traditionally thought to have reduced genetic variation relative to their source populations because of genetic founder effects linked to small population size during the introduction and establishment phases of an invasion [66]. Furthermore, the genetic variability found clustered according to geographical ranges is in contrast with the possibility that *B. vicetinus* may be an invasive species accidentally introduced.

Even though outbreaks have been reported on both the Euganean and Berici Hills, different haplotypes have been found in these two distribution ranges. Ninety-four percent of haplotypes were exclusive to a single distribution range (i.e. either Euganean or Berici Hills) and only one haplotype, scarcely represented, was found in samples from both hilly areas. These results suggest that outbreaks are not a consequence of a single or few pestiferous haplotypes but rather that the source of outbreaks is due to local populations which experienced a demographic increase in each area. Thus, it seems that outbreaks have appeared independently from the genetic origin, as also found in some studies which indicate that outbreak events are often more affected by environmental conditions than by genetic characteristics of the local population [4, 67–69].

The geographical separation between populations of the Euganean and Berici Hills, observed in the network and confirmed by population differentiation test and AMOVA, indicates a limited gene flow among populations. In phytophagous insects, dispersal capacity, geographical or reproductive barriers, host plant, and habitat fragmentation are reported as the main drivers of genetic structure [20, 70, 71]. The limited dispersal ability of this flightless species could have favoured the lack of gene flow among its distribution ranges. *B. vicetinus* has been reported to show a low dispersal ability [9], as well as other ground-dispersing species that move only relatively short distances, such as 100–200m during their whole life (e.g. *Pholidoptera griseoaptera*, [72]). In addition, the mostly lowland areas between the Euganean and Berici Hills, with the presence of agricultural fields and the absence of woody vegetation, might be hostile areas for bush-cricket survival and could have acted as a geographical barrier limiting the effective dispersal of the species. Furthermore, spatial configuration of habitats is another factor influencing genetic structure, that affects mainly species with limited dispersal ability [73]. In *B. vicetinus* outbreak areas, it has been observed that habitat loss and the presence of patchy areas of the non-host alien tree *Robinia pseudoacacia* play an important role in reducing population density and dispersion [9]. These factors, coupled to the low mobility of the pest, could have a synergic action that might explain the high level of differentiation observed among populations across its distribution ranges.

High haplotypic diversity, low levels of sequence divergence (nucleotide diversity) and a star-like phylogeny were observed in populations of *B. vicetinus* from the Euganean Hills and to a lesser extent from the Berici Hills (Fig 1; Table 1). This pattern is consistent with what is expected for populations that have experienced past demographic expansions [52, 74]. These results were highlighted by the neutrality tests and the unimodal mismatch distribution in both hilly areas supporting that, besides the current and ongoing outbreaks, populations from the Euganean and Berici Hills underwent an expansion after a period of low effective sample size (S4 Fig). The bush-cricket population of the Euganean Hills experienced a postglacial expansion starting approximately 10,700–12,400 years ago after the Last Glacial Maximum (LGM; 21,000 years ago) whereas the expansion process in populations on the Berici Hills could have occurred at the end of the LGM. During this glacial period, climate effects were minimal in the Euganean Hills, with thermophilic vegetation serving as a refuge for several species [75, 76]. Accordingly, *B. vicetinus* populations could have survived during the climatic oscillations, and exploited these hilly areas with potentially suitable environmental conditions

throughout the late Pleistocene. Once climatic conditions were favourable, at the end of the last ice age, populations might have experienced an expansion, shaping the present genetic structure of the species. The slow movement of the species and lack of host plants in the low-lands during these periods probably prevented dispersal of the species between the two hilly areas, favouring differentiation of the mitochondrial haplotypes. Thus, both historical (e.g., expansion after LGM) and recent changes (e.g. habitat loss) have contributed to determining the genetic structure and diversity of the bush-cricket in its outbreak range, resulting in geneti-cally structured populations.

Further studies increasing the sampling size, mostly in areas where the species is rare (e.g. Lessini Mountains), combined with analysis of other markers could help obtain a better pic-ture of *B. vicetinus* population structure.

The current findings contribute to a growing knowledge of the status and evolutionary his-tory of the pest. Here, by removing doubt about the origin of the *B. vicetinus*, we have achieved an important step in understanding this native species. Shedding light on the origin of a spe-cies is also important for its biological control. Given that natural enemies and their host tend to coevolve, biological control programmes often rely on the use of parasitoids present in the native areas. Failure to identify the correct origin of a pest may lead to the use of unsuitable species as biocontrol agents with negative effects on control programmes [20, 77]. Maximizing efforts to locate native natural enemies (native parasitoids), as the egg parasitoid reported in Ortis et al. [13] will be useful for control programmes. Finally, although *B. vicetinus* is cur-rently in outbreak status, control programme strategies should consider the intrinsic vulnera-bility of an endemic species. Its low dispersal ability and small and fragmented distribution range could lead this native species to quickly shift from outbreaking to endangered.

## Supporting information

**S1 Fig. *Barbitistes vicetinus* specimens and damage.**
(TIF)

**S2 Fig. Parsimony network and geographic distribution of *Barbitistes vicetinus* haplotypes.**
Network obtained with the COI-tRNALeu-COII data set.
(PDF)

**S3 Fig. Parsimony network and geographic distribution of *Barbitistes vicetinus* haplotypes.**
Network obtained with the COI-tRNALeu-COII data set. B) Network obtained with the
12S-CR data set. Network obtained with the 12S-CR data set.
(PDF)

**S4 Fig. Mismatch distribution under the population expansion model of *Barbitistes viceti-nus* populations in the disjunct ranges.**
(PDF)

## Acknowledgments

We thank Elena Cappelli and Nicolo Scarabottolo for their help in collecting and analysing samples.

## Author Contributions

**Conceptualization:** Isabel Martinez-Sañudo, Corrado Perin, Luca Mazzon.

**Data curation:** Isabel Martinez-Sañudo, Corrado Perin.

**Formal analysis:** Isabel Martinez-Sañudo, Corrado Perin.

**Funding acquisition:** Luca Mazzon.

**Investigation:** Isabel Martinez-Sañudo, Corrado Perin, Giacomo Cavaletto, Giacomo Ortis.

**Methodology:** Isabel Martinez-Sañudo, Corrado Perin, Giacomo Cavaletto, Giacomo Ortis, Luca Mazzon.

**Project administration:** Luca Mazzon.

**Resources:** Corrado Perin, Giacomo Cavaletto, Giacomo Ortis, Paolo Fontana.

**Supervision:** Isabel Martinez-Sañudo, Luca Mazzon.

**Validation:** Luca Mazzon.

**Writing – original draft:** Isabel Martinez-Sañudo, Corrado Perin, Luca Mazzon.

**Writing – review & editing:** Isabel Martinez-Sañudo, Corrado Perin, Giacomo Cavaletto, Giacomo Ortis, Paolo Fontana, Luca Mazzon.

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
