## [Decision Letter · Decision Letter 0]

1 Mar 2021

PONE-D-20-40749

Studying genetic population structure to shed light on the demographic explosion of the rare species Barbitistes vicetinus (Orthoptera, Tettigoniidae)

PLOS ONE

Dear Dr. Martinez-Sañudo,

Thank you for submitting your manuscript to PLOS ONE. After careful consideration, we feel that it has merit but does not fully meet PLOS ONE’s publication criteria as it currently stands. Therefore, we invite you to submit a revised version of the manuscript that addresses the points raised during the review process.

We look forward to receiving your revised manuscript.

Kind regards,

Bi-Song Yue, Ph.D

Academic Editor

PLOS ONE

Journal Requirements:

"This research was partially supported by the project DOR1881327/18 - University of Padua to Luca Mazzon. The funders had no role in study design, data collection and analysis, decision to publish, or preparation of the manuscript."

3. We note that Figure 1 in your submission contain map images which may be copyrighted. All PLOS content is published under the Creative Commons Attribution License (CC BY 4.0), which means that the manuscript, images, and Supporting Information files will be freely available online, and any third party is permitted to access, download, copy, distribute, and use these materials in any way, even commercially, with proper attribution. For these reasons, we cannot publish previously copyrighted maps or satellite images created using proprietary data, such as Google software (Google Maps, Street View, and Earth). For more information, see our copyright guidelines: http://journals.plos.org/plosone/s/licenses-and-copyright.

3.1.    You may seek permission from the original copyright holder of Figure 1 to publish the content specifically under the CC BY 4.0 license. 

3.2.    If you are unable to obtain permission from the original copyright holder to publish these figures under the CC BY 4.0 license or if the copyright holder’s requirements are incompatible with the CC BY 4.0 license, please either i) remove the figure or ii) supply a replacement figure that complies with the CC BY 4.0 license. Please check copyright information on all replacement figures and update the figure caption with source information. If applicable, please specify in the figure caption text when a figure is similar but not identical to the original image and is therefore for illustrative purposes only.

Reviewers' comments:

Reviewer's Responses to Questions

**Comments to the Author**

1. Is the manuscript technically sound, and do the data support the conclusions?

Reviewer #1: Yes

Reviewer #2: Yes

2. Has the statistical analysis been performed appropriately and rigorously? 

Reviewer #1: Yes

Reviewer #2: Yes

3. Have the authors made all data underlying the findings in their manuscript fully available?

Reviewer #1: No

Reviewer #2: Yes

4. Is the manuscript presented in an intelligible fashion and written in standard English?

Reviewer #1: Yes

Reviewer #2: Yes

5. Review Comments to the Author

Reviewer #1: he authors present analyses of mtDNA variation intended to shed light on the origin of a recent pest insect outbreak. The analyses are sound as are the conclusions. I have two main comments and a few minor issues outlined below.

line 169 I think this paragraph should be deleted since the tree is not mentioned (maybe on line 379?) or presented anywhere. Nor is it needed given that assumptions of bifurcating relationships is likely violated (which is presumably the justification for the TCS network).

lines 309-313 One major issue seems to be neglected: how do the estimates of the timing of demographic change compare to the ideas presented in the introduction that the change has occurred very recently? The inferences from molecular data indicate a post-Pleistocene time frame for demographic expansion, but the introduction (line 54) indicates the outbreak has occurred over the last decade. As presented, the attempt to time the demographic events seems to indicate that the authors are thinking of the very recent outbreak, though I suspect, given the nature of the data and the analyses, that only historical events can be detected. I suggest the authors re-arrange the questions and methods to separate these issues and to be more clear on why they are interested in timing the demographic events (which don't have anything to do with the last decade). Temporal sampling with 100's to 1000's of bi-parentally inherited markers might be required to detect changes in Ne over the very near term.

minor comments

line 41 this is not clear - what is meant by "genetic origin"?

line 44 how does mtDNA help with biocontrol strategies? This idea could be expanded by including some of the logic that appears in the discussion.

line 52 I think that some more precise language is needed rather than "ecological balance", a phrase that has had a controversial and tortuous history in ecology.

line 97 DO the authors have the requiste geographical sampling to answer this question of alien origin?? It seems that some basis for comparisons is required for this.

line 114 Change "expositions" to "exposures"

lines 106-121 refer to map fig 1

line 169 I think this paragraph should be deleted since the tree is not presented and not mentioned (maybe on line 379?) or presented anywhere. Nor is it needed given that assumptions of bifurcating relationships is likely violated (which is presumably the justification for the TCS network)?

line 181 delete "a" before coalescent

line 192 delete "occurred"

line 216 it might be useful to describe what you are looking for, i.e. what distinguishes numts from mtDNA genes?

line 229 delete the percent sign

line 379 what aspects are unresolved? The data presented seem to me to provide a fairly clear picture. Further, why "phylogenetic" resolution? The network clearly demonstrates shared haplotypes indicative of gene flow or, more likely, incomplete lineage sorting for which phylogenetic methods are inappropriate - see comment above (given assumption in phylogenetic methods of bifurcating relationships and extinct ancestral states).

FIg 1 top: what do the lines within haplotype circles signify? I think the authors might be indicating proportions shared between regions (haplotypes A1 and I1) or among subregions (i.e. A6, I5) but this is not clear from the colouration.

Reviewer #2: A great application of these techniques. It would be nice to see similar application to other pest species.

I have only minor questions:

(1) How significant is the limited geographical mobility to your interpretation? Would you not come to similar conclusions on species that can disperse but usually show site fidelity?

(2) I would like to see some discussion of application to other species.

(3) What light can such techniques shed on the efficacy of geographically-based pest control methods (such as quarantines)?

6. PLOS authors have the option to publish the peer review history of their article (what does this mean?). If published, this will include your full peer review and any attached files.

Reviewer #1: No

Reviewer #2: **Yes: **Shripad Tuljapurkar

---

## [Author Response · Author response to Decision Letter 0]

25 Mar 2021

Reviewer #1 

The authors present analyses of mtDNA variation intended to shed light on the origin of a recent pest insect outbreak. The analyses are sound as are the conclusions. I have two main comments and a few minor issues outlined below.

line 169 I think this paragraph should be deleted since the tree is not mentioned (maybe on line 379?) or presented anywhere. Nor is it needed given that assumptions of bifurcating relationships is likely violated (which is presumably the justification for the TCS network).

Ok. Thank you. We deleted this paragraph.

lines 309-313 One major issue seems to be neglected: how do the estimates of the timing of demographic change compare to the ideas presented in the introduction that the change has occurred very recently? The inferences from molecular data indicate a post-Pleistocene time frame for demographic expansion, but the introduction (line 54) indicates the outbreak has occurred over the last decade. As presented, the attempt to time the demographic events seems to indicate that the authors are thinking of the very recent outbreak, though I suspect, given the nature of the data and the analyses, that only historical events can be detected. I suggest the authors re-arrange the questions and methods to separate these issues and to be more clear on why they are interested in timing the demographic events (which don't have anything to do with the last decade). Temporal sampling with 100's to 1000's of bi-parentally inherited markers might be required to detect changes in Ne over the very near term.

The analyses of past demographic events were conducted to understand how past events could have shaped the genetic structure of B. vicetinus populations. The time since the expansion events revealed by the analysis was calculated to infer the historical period in which the expansion could have taken place, being aware that this expansion is different from the recent outbreak events. However, this analysis is secondary in importance compared to the genetic diversity analysis. 

To be clearer and avoiding misunderstandings we delete one sentence in the abstract, modified the third question in the introduction (line 101), as well as some sentences in discussion (lines 366-368), paying more attention to keep separated these issues.

line 41 this is not clear - what is meant by "genetic origin"? We changed this unclear word. Please see line 40

line 44 how does mtDNA help with biocontrol strategies? This idea could be expanded by including some of the logic that appears in the discussion. Thank you for your suggestion. We briefly expanded it. Please see lines 43-44.

line 52 I think that some more precise language is needed rather than "ecological balance", a phrase that has had a controversial and tortuous history in ecology. Done. Please see lines 52.

line 97 DO the authors have the requiste geographical sampling to answer this question of alien origin?? It seems that some basis for comparisons is required for this. We agree with the reviewer. However, the distribution range of B. vicetinus is limited to the small area of north-east Italy where the sampling was performed. The species has not been reported elsewhere, so far.

In addition, we have conducted a preliminary phylogenetic study of the species belonging to the Western-Palearctic genus Barbitistes (through barcoding). Sequences of B. vicetinus were different from sequences of other Barbitistes sp. and all the B. vicetinus specimens grouped in a monophyletic clade (unpublished data). 

We added more details in the text. Please see line 67-68.

line 114 Change "expositions" to "exposures” Done. Please line 115.

lines 106-121 refer to map fig 1 Done. Please line 111.

line 169 I think this paragraph should be deleted since the tree is not presented and not mentioned (maybe on line 379?) or presented anywhere. Nor is it needed given that assumptions of bifurcating relationships is likely violated (which is presumably the justification for the TCS network)? Done.

line 181 delete "a" before coalescent Done. Please see line 179.

line 192 delete "occurred" Thank you. Done. See line 190.

line 216 it might be useful to describe what you are looking for, i.e. what distinguishes numts from mtDNA genes? Ok. Thank you. We added more information. See line 214-218.

line 229 delete the percent sign Done. Please see line 232.

line 379 what aspects are unresolved? The data presented seem to me to provide a fairly clear picture. Further, why "phylogenetic" resolution? The network clearly demonstrates shared haplotypes indicative of gene flow or, more likely, incomplete lineage sorting for which phylogenetic methods are inappropriate - see comment above (given assumption in phylogenetic methods of bifurcating relationships and extinct ancestral states). We agree with the reviewer. We modified it. Please see line 384-385.

FIg 1 top: what do the lines within haplotype circles signify? I think the authors might be indicating proportions shared between regions (haplotypes A1 and I1) or among subregions (i.e. A6, I5) but this is not clear from the colouration. Thank you for your suggestion

We added this information in the figure legend trying to make it clearer:

Lines within haplotypes circles indicate the proportions shared between collection areas

(Please see lines 263-264) 

Reviewer #2 

A great application of these techniques. It would be nice to see similar application to other pest species.

I have only minor questions:

(1) How significant is the limited geographical mobility to your interpretation? Barbitistes vicetinus tends to stay close to its hatching site during the whole life cycle, likely due to its low dispersal ability. This characteristic, coupled to the low density of the species until 2008, probably helped to maintain the strong genetic structure of the species. However, in correspondence with the outbreaks, a tendency of the insect to disperse more than before was reported (Cavaletto et al. 2018). During the outbreaks, B. vicetinus have colonized new habitats and moved far from the hatching areas, perhaps to avoid intraspecific competition. 

Besides the low mobility of the bush-cricket other factors such as geographical barriers and spatial configuration could have favoured the absence of gene flow between B. vicetinus populations. The lowland areas between the Euganean and Berici Hills, with several agricultural fields and without woody vegetation could have acted as a geographical barrier, notwithstanding the short geographical distance. We have also observed that patchy areas of the non-host alien tree Robinia pseudoacacia have played an important role in reducing population density and dispersion of the pest (Cavaletto et al. 2019). 

(1) b) Would you not come to similar conclusions on species that can disperse but usually show site fidelity? The low dispersal ability is undoubtedly a contributing factor to maintaining a high levels of genetic differentiation. Other species that have the ability to move but show fidelity to their life site could also show a strong genetic structure, although it may be less marked as with low-mobility species. Species showing breeding site fidelity could move if necessary (e.g. habitat loss, climate changes) and it could modify the genetic structure.

(2) I would like to see some discussion of application to other species. Similar studies can be applied to a wide range of insect species. Population genetic strategies have been widely used for examining patterns and magnitude of dispersal of several insects over geographic and temporal scales. Regarding invasive species, for example, population genetics studies can be conducted for highlighting and discovering the sources and colonization routes of these alien insects. 

The information obtained with genetic analyses can be also useful for an effective conservation of endangered species. Indeed, analysing the genetic diversity patterns of endangered species has become an integral component of many management strategies. 

Some examples of the application of these techniques to other species can be found in lines 74-83 and 93-95.

(3) What light can such techniques shed on the efficacy of geographically-based pest control methods (such as quarantines)? The genetic techniques used in this study have allowed us to realize that population outbreaks are not always a consequence of a single or few pestiferous haplotypes. As reported here, any individual of this species, regardless of its genetic haplotype, had an equal chance of producing outbreaks. It follows that control strategies must always be applied (e.g.: strategies based on geographic control). 

For species characterized by a reduced movement behaviour throughout their lifespan (such as the bush-cricket here studied), gene flow appears low enough to maintain a high genetic structure among populations. In this case, control programs based on isolated treatments (i.e. at a small geographic scale) could be also effective. 

Thus, these results based on genetic techniques strengthen the need to apply and enhance control methods such as quarantines or other strategies based on geographic control.

---

## [Editor Report · Decision Letter 1]

8 Apr 2021

Studying genetic population structure to shed light on the demographic explosion of the rare species Barbitistes vicetinus (Orthoptera, Tettigoniidae)

PONE-D-20-40749R1

Dear Dr. Martinez-Sañudo,

We’re pleased to inform you that your manuscript has been judged scientifically suitable for publication and will be formally accepted for publication once it meets all outstanding technical requirements.

Kind regards,

Bi-Song Yue, Ph.D

Academic Editor

PLOS ONE

---

## [Editor Report · Acceptance letter]

22 Apr 2021

PONE-D-20-40749R1 

Studying genetic population structure to shed light on the demographic explosion of the rare species *Barbitistes vicetinus* (Orthoptera, Tettigoniidae) 

Dear Dr. Martinez-Sañudo:

I'm pleased to inform you that your manuscript has been deemed suitable for publication in PLOS ONE. Congratulations! Your manuscript is now with our production department. 

Kind regards, 

on behalf of

Dr. Bi-Song Yue 

Academic Editor

PLOS ONE